# Intermediate Layers Matter in Momentum Contrastive Self Supervised Learning

**Aakash Kaku**
Center for Data Science
New York University
New York, NY 10011
ark576@nyu.edu

**Sahana Upadhya**
Department of Computer Science
Courant Institute of Mathematical Sciences
New York, NY 10011
su575@nyu.edu

**Narges Razavian**
Departments of Population Health and Radiology
NYU Grossman School of Medicine
and NYU Center for Data Science
New York, NY 10016
Narges.Razavian@nyulangone.org

## Abstract

We show that bringing intermediate layers' representations of two augmented versions of an image closer together in self supervised learning helps to improve the momentum contrastive (MoCo) method. To this end, in addition to the contrastive loss, we minimize the mean squared error between the intermediate layer representations or make their cross-correlation matrix closer to an identity matrix. Both loss objectives either outperform standard MoCo, or achieve similar performances on three diverse medical imaging datasets: NIH-Chest Xrays, Breast Cancer Histopathology, and Diabetic Retinopathy. The gains of the improved MoCo are especially large in a low-labeled data regime (e.g. 1% labeled data) with an average gain of 5% across three datasets. We analyze the models trained using our novel approach via feature similarity analysis and layer-wise probing. Our analysis reveals that models trained via our approach have higher feature reuse compared to a standard MoCo and learn informative features earlier in the network. Finally, by comparing the output probability distribution of models fine tuned on small versus large labeled data, we conclude that our proposed method of pre-training leads to lower Kolmogorov–Smirnov distance, as compared to a standard MoCo. This provides additional evidence that our proposed method learns more informative features in the pre-training phase which could be leveraged in a low-labeled data regime.

## 1 Introduction

Self-supervised learning (SSL) [Noroozi and Favaro, 2016, Doersch et al., 2015, Zhang et al., 2016, He et al., 2020, Chen et al., 2020, Caron et al., 2020, Zbontar et al., 2021, Caron et al., 2021] helps in learning useful representations from large unlabeled dataset that can be used as pre-trained initialization points for different downstream tasks. These pretrained models are particularly useful in settings where we have a small labeled dataset to learn from. Recent advances have shown that such self-supervised methods are capable of learning representations that are competitive with those learned in a fully supervised manner for natural images [Wu et al., 2018, Hjelm et al., 2018, Zbontar et al., 2021, Caron et al., 2021]. In application domains such as medical imaging, where high quality labeled data is scarce and data modalities shift rapidly, SSL has potential to make a significant impact.

35th Conference on Neural Information Processing Systems (NeurIPS 2021).

Contrastive self-supervised learning methods [He et al., 2020, Chen et al., 2020, Caron et al., 2020, Zbontar et al., 2021, Caron et al., 2021] are currently the state of the art in SSL. In these methods, final layer's feature representations for positive pairs of images are brought closer and those for negative pairs are pushed away using a contrastive loss functions such as InfoNCE [Oord et al., 2018] or NT-Xent [Chen et al., 2020]. A positive pair of image could be two different views of the same image while a negative pair could be views of different images. These contrastive methods have also been applied to various medical imaging tasks such as chest x-ray interpretation [Sowrirajan et al., 2021, Vu et al., 2021], image segmentation [Chaitanya et al., 2020], covid diagnosis [Chen et al., 2021] and dermatology classification [Azizi et al., 2021]

In this work, we extend the idea of contrastive learning further and hypothesize that ensuring closeness of representations for the intermediate layers, and not just the last one, significantly improves the learning of network parameters. To this end, we hypothesize that the features learned by the method should be more reusable for the downstream task than a standard contrastive learning method. We use Momentum Contrastive (MoCo) method [He et al., 2020] as the base SSL method, and study two additional loss terms based on closeness of feature representation between intermediate layers. The first loss function is based on mean squared error, and the second loss function is based on distance between cross-correlation matrix and identity matrix [Zbontar et al., 2021].

We test our method on three diverse medical datasets:NIH chest x-ray, diabetic retinopathy and breast cancer histopathology. We show that both proposed loss functions either outperform standard MoCo or have comparable performance for classification tasks on all three datasets. The performance gap between the standard MoCo and our approach is higher in the case where we have a small labeled dataset to fine tune our model. This suggests that the features learned via our self-supervision approach are of higher quality than the standard MoCo. To confirm this finding, we first investigate how much the features are *reused* by the model, by comparing the similarity of the features before and after fine-tuning on the labeled dataset. Based on prior work [Neyshabur et al., 2021], higher feature reuse indicates better feature quality. Additionally, we also perform layer-wise probing of the model to conclude that indeed the features learned earlier in the model by our approach are more informative than a standard MoCo. Finally, to understand the importance of the feature quality for learning from a small labeled dataset, we compare the output probability distribution of models fine tuned on 1% and 6% of labeled data versus on 100% of labeled data. The proposed method of pre-training leads to lower Kolomongrov-Smrinov (KS) distance against the best performing model, as compared to a standard MoCo. To the best of our knowledge, this is the first work to show that the proposed approach has high performance on such diverse non-natural imaging datasets. Additionally, this is the first work to investigate the feature quality of the self-supervised method by going beyond the linear probing and focusing on feature reuse and on learning informative features earlier in the model.

## 2 Relevant work

### 2.1 Pretext Task based SSL

Self-supervised learning helps to learn feature representation of data in an unsupervised manner. This is accomplished by performing a formulated task where the labels for the task are extracted from the data itself. Classic examples for handcrafted formulated tasks include solving a jigsaws puzzle [Noroozi and Favaro, 2016], relative patch prediction [Doersch et al., 2015] and colorization [Zhang et al., 2016]. These tasks are based on heuristics and therefore the representations learned using these tasks may not be useful for performing downstream tasks. Consequently, contrastive methods to learn representations in an unsupervised manner emerged. They outperformed the heuristics task based methods and gradually have also reached a fully supervised level of performance [Caron et al., 2021]. Some of the recent contrastive methods include SimCLR [Chen et al., 2020], MoCo [He et al., 2020], BYOL [Grill et al., 2020], SwAV [Caron et al., 2020], Barlow Twins [Zbontar et al., 2021] and Dino [Caron et al., 2021].

All the contrastive methods treat the backbone model as a black box and work with the final layer's feature representations only. A recent work [Chengyue Gong, 2021] attempted to incorporate contrastive loss for the intermediate layers of a model trained via MoCo, with a focus on accelerating the model training. They employed partial back-propagation - by randomly choosing a layer to back-propagate from - for reducing model computations and training time. In contrast, our work focuses on investigating the impact of the intermediate loss terms on feature quality, using state of the art analysis methods such as feature re-use, layer-wise probing and model output similarity. We

investigate this intermediate loss functions with two distinct formulations (mean squared error, and a loss based on cross correlation matrix) and on three diverse medical imaging datasets. We do not employ partial back propagation, and our formulation of intermediate layer loss does not require storing intermediate representations for the negative pairs.

## 2.2 Contrastive self supervised learning in medical imaging

Out of above mentioned contrastive methods, MoCo has been widely adopted by the research community across many domains including medical imaging. The momentum contrastive method (MoCo) differentiates from other contrastive methods by having an additional memory bank of negative examples. The memory bank helps the model to learn even when the batch size is small, and is especially important for medical imaging tasks, because medical images are typically large (e.g. $1024 \times 1024$) and it becomes computationally infeasible to train with a large batch size. Increasingly, many recent works have used momentum contrastive method with minor modifications as the choice of self-supervised method to perform diverse medical imaging tasks such as histopathology classification [Ciga et al., 2020, Dehaene et al., 2020], chest x-ray interpretation [Sowrirajan et al., 2021] and dermatology classification [Azizi et al., 2021]. Given a widespread adoption of the momentum contrastive method, we focus on it in our work.

# 3 Methodology

In section 3.1, first, we briefly discuss the standard MoCo method. Then, we describe our proposed approach in section 3.2. In section 3.3, we describe the fine tuning and evaluation methodology for the self-supervised methods. Finally, in section 3.4, we describe the methods used for feature analysis which helps us quantify the quality of the representations learned by different methods.

## 3.1 Standard MoCo

With the help of MoCo, we want to learn feature representation for images in an unsupervised manner. We achieve this by minimizing InfoNCE loss [Oord et al., 2018]

$$\mathcal{L}_{InfoNCE}(x) = -log \frac{\exp(g(\tilde{x_1}).h(\tilde{x_2})/\tau)}{\exp(g(\tilde{x_1}).h(\tilde{x_2})/\tau) + \sum_{i=0}^{K} \exp(g(\tilde{x_1}).h(\tilde{z_i})/\tau)} \tag{1}$$

where $\tilde{x_1}$ and $\tilde{x_2}$ are different views/augmentations of the same image $x$. $\tilde{x_1}$ and $\tilde{x_2}$ are called a positive pair. There are $K$ negative pairs of $\tilde{x_1}$ and $\tilde{z_i}$s where $\tilde{z_i}$ is random augmentation of a different image. $\tilde{z_i}$ either comes from a queue in the case of MoCo or a minibatch in the case of SimCLR [Chen et al., 2020]. We minimize equation 1 to optimize the parameters of the encoder $g$. $h$ is an identical copy of $g$ whose parameters are an exponential moving average of $g$. $\tau$ is a temperature-scaling hyper-parameter.

## 3.2 MoCo with intermediate feature similarity

In our work, we extend the idea of contrastive learning and propose that the encoder should be encouraged to learn augmentation-invariant representations not only at the end of the encoder but also for intermediate layers. We make use of an additional loss function to ensure closeness in the representations for the intermediate layers. Such intermediate loss have shown to improve performance of deep learning models such as InceptionNet [Szegedy et al., 2015], PSPnet [Zhao et al., 2017] and DARTs [Liu et al., 2018]. Intermediate loss helps to improve performance for various reasons. In Mostajabi et al. [2018], it helped as a regularizer. In InceptionNet, PSPnet and DARTs, it improved training of the model by facilitating gradient flow. In our case, we hypothesize that it would facilitate learning of augmentation-invariant features early in the model thereby contributing to learning of high quality features for the downstream tasks.

We experiment with two intermediate loss functions that ensure closeness of representations: mean square error, and a loss function inspired by Zbontar et al. [2021] that minimizes the difference between cross-correlation matrix of representations and an identity matrix. Figure 3.2 includes an overview of our approach.

### 3.2.1 Mean squared error of intermediate features

We compute the mean squared error of the intermediate features between the two views of the same image. We extract the intermediate features and pass them through an adaptive average pooling layer

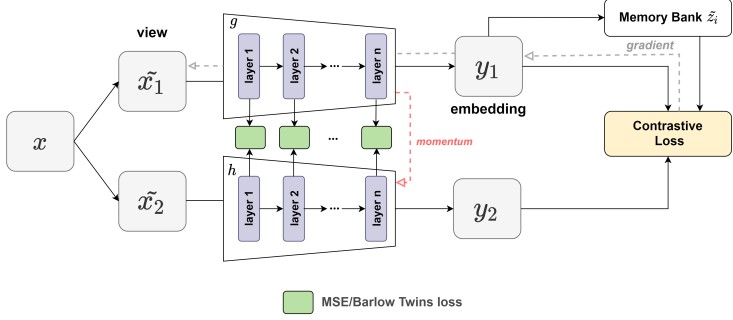

Figure 1: Schematic diagram for our approach of self-supervised learning. In addition to a contrastive loss, we use intermediate loss like mean squared error (MSE) or Barlow twins loss to bring intermediate representations of the two views of same image closer together. The "momentum" in the figure denotes that the weights of the second encoder are an exponential moving average of the weights of the first encoder.

to reduce the spatial variances in the two views that may be caused due to augmentations such as rotations. For ResNet-50 as the backbone architecture, the spatial resolutions of the features after the average pooling layers are $16 \times 16$, $16 \times 16$, $4 \times 4$ and $4 \times 4$ for "Block 1", "Block 2", "Block 3" and "Block 4" of ResNet-50, respectively. The mean squared error is calculated between these spatially down-sampled features of the two views for each block.

### 3.2.2 Cross-correlation based loss function for intermediate features

We took inspiration from Zbontar et al. [2021] for this loss function. The loss function is termed *Barlow Twins (BT)*, and is designed to bring the cross-correlation matrix of two feature representations close to an identity matrix. The loss function is defined as follows:

$$\mathcal{L}_{\mathcal{BT}} = \underbrace{\sum_{i}(1 - \mathcal{C}_{ii})^2}_{\text{invariance term}} + \ \lambda \ \underbrace{\sum_{i}\sum_{j \neq i} \mathcal{C}_{ij}^2}_{\text{redundancy reduction term}} \qquad \mathcal{C}_{ij} = \frac{\sum_b z_{b,i}^A z_{b,j}^B}{\sqrt{\sum_b \left(z_{b,i}^A\right)^2} \sqrt{\sum_b \left(z_{b,j}^B\right)^2}}$$

where, $\lambda$ is a weight that trades off between the *invariance term* and *redundancy term*. Each $\mathcal{C} \in [-1, 1]^{d \times d}$ is the cross-correlation matrix of two feature representations of $d$ dimension. Here, $z^A$ and $z^B$ are a batch of intermediate feature representations for two augmented versions of the same image with batch size $b$. $i$ and $j$ are feature components of $d$ dimensional feature.

The *invariance term* tries to equate the diagonal entries of the correlation matrix to one, thereby making the representations invariant to augmentations. The *redundancy term* tries to equate the non-diagonal terms to zero and thereby decorrelating the different components of the feature representation. This decorrelation helps to reduce redundancy encoded in different components of the representation.

We follow Zbontar et al. [2021]'s procedure for implementing the Barlow twins loss function. We pass the intermediate feature representation of dimension $\mathbb{R}^{C \times H \times W}$ through a global average pooling and obtain a feature vector of $\mathbb{R}^C$ dimension. Here, $C$ = number of channels, $H$ = height of the each channel, and $W$ = width of each channel. The resultant feature vector is passed through three linear layers with ouput dimension being 2048 for each linear layer. The first two linear layers are followed by ReLU non-linearity and batch normalization. The resultant output representations are fed to the Barlow twins loss function.

Intermediate feature similarity applies only to positive pairs. For both loss functions, we do not need negative pairs to learn the representations. We, therefore, do not have to store the intermediate representations for the negative pairs which could be computationally expensive.

### 3.3 Model fine tuning methodology

In order to evaluate the utility of the representations learned by models using our approach and the standard MoCo, we use the pre-trained models to perform downstream classification tasks. While fine tuning the models for performing the downstream tasks, we perform two types of fine tuning: fine tuning only the final linear classifier (LL-fine-tuning) or fine tuning the entire model (E2E

fine-tuning). We fine tune the model using different fraction of labeled training data i.e. 1%, 6% and 100%. For example, if a model is fine tuned with 1% label fraction, the model will have access to only 1% of the labels of the training data which was used for self-supervised pre-training of the model. We ensured that the class distribution of the 1% and 6% label fraction matches the 100% label fraction training data (see Appendix D). Fine tuned model with best validation performance is used for obtaining predictions for test set. We report all the performance measures with 95% confidence interval which we calculate using 1000 bootstrap replicates of the test set. We calculate the upper and lower confidence limit using $\mu \pm 1.96\sigma/\sqrt{N}$ where $\mu$ and $\sigma$ are mean and standard deviation of the performance measure for 1000 bootstrap replicates, respectively. $N$ is number of bootstrap replicates.

### 3.4 Analysis of the features

We analyze the features of the models learned via self-supervised methods to understand their quality for the downstream tasks. To objectively measure the quality, we investigate the re-usability of the features. Additionally, we also probe the pre-trained models in a layer-wise manner to understand how informative the intermediate features for the downstream tasks are. Finally, we compute the KS distance between the distributions of output probability of the model fine tuned on small vs large data. This shows us the utility of different self-supervised methods when there is a small labeled dataset available for learning the downstream task.

#### 3.4.1 Feature reuse

Higher feature reuse indicates better feature quality [Neyshabur et al., 2021]. To assess feature reuse in SSL, we measure the similarity of the features of the models trained by our proposed method, before and after fine tuning them on a labeled dataset. We compute the similarity of features extracted from different layers of the pre-trained and E2E-fine-tuned models. Intuitively, if two models pre-trained via self-supervised learning reach a similar performance after fine-tuning on a downstream task, the SSL method that leads to the model with features closer to the fine-tuned model has learned better/more-useful features during the self supervised learning.

Following Neyshabur et al. [2021], we use Centered Kernal Alignment (CKA) [Kornblith et al., 2019] as the measure of feature similarity. CKA has desirable properties such as invariance to invertible linear transformations, orthogonal transformations and isotopic scaling. We use CKA with RBF kernel and $\sigma = 0.8$ following prior work. To compute the feature similarity, we extract the intermediate features from the model which has a dimensions of $\mathbb{R}^{C \times H \times W}$ where, $C$ = number of channels, $H$ and $W$ are height and width of each channel. We pass each channel through a global average pooling layer and use the resultant vector of $\mathbb{R}^C$ dimension for computing feature similarity.

#### 3.4.2 Layer-wise probing

We probe the intermediate layers of the pre-trained models to understand how informative the intermediate features are for performing the downstream tasks. We expect the intermediate features learned via our approach to be more informative than the standard MoCo, because by bringing the intermediate features of two augmented version of an image closer together, our approach encourages the model to learn augmentation-invariant features not only late in the network but at all levels.

**Procedure for layer-wise probing**: We extract the intermediate features from the model which has a dimensions of $\mathbb{R}^{C \times H \times W}$. We pass it through a global average pooling layer and use the resultant feature vector of size $\mathbb{R}^C$ for performing the downstream task. The feature vector is finally passed through a linear classifier. The entire network is frozen except the final linear classifier. Since all our experiments are performed on ResNet-50 as the backbone architecture, we extract features at the end of each ResNet block namely, "Block 1", "Block 2", "Block 3" and "Block 4".

#### 3.4.3 Feature utility for small datasets

In medical domain, we typically encounter a small labeled dataset from which we want to learn a generalizable model. This is often challenging and in such scenarios, self-supervised pre-trained models can be useful. In this analysis, we investigate how close a self-supervised model, fine tuned on a small labeled dataset, gets to one that utilizes a larger labeled dataset. To this end, we compare the distributions of the output probability of models, fine tuned on small versus large labeled data using Kolmogorov-Smirnov (KS) distance [Massey Jr, 1951]. KS distance between two distributions is the greatest separation between their cumulative distribution functions (CDFs). We compute KS distance between models trained via our SSL methods, fine tuned on 1% and 6% of the labeled training data, and the best model fine-tuned on 100% of the labeled training data (Detailed procedure

to compute KS distance is described in Appendix G). KS distance not only depends on the difference in the performance of the two models but also on how similarly/differently calibrated the two models are [Gupta et al., 2020]. Therefore, this metric goes beyond the standard linear probing evaluation as it also takes into account the model calibration. Lower KS distance indicates a better method for self supervised learning, when only a small labeled dataset is available.

## 4 Experiment and Results

### 4.1 Experimental settings

We use Resnet-50 as the backbone architecture for all the baseline self-supervised methods, supervised models and our approach. We use pytorch-lightning's [Falcon et al., 2019] implementation of MoCo as the basis for our analysis. A preliminary ablation analysis showed that applying intermediate loss to all blocks yielded superior results (See Appendix F). Therefore, for our approach, we apply the intermediate feature loss after each of the four ResNet blocks. Specifically, the four intermediate feature losses from each block of the Resnet-50 encoder are added to the InfoNCE loss using a scaling factor. The scaling factor multiplies the intermediate feature loss by 0.25 (in case of MSE loss) or by 5e-5 (in case of BT loss). We chose these scaling factors so that the contrastive loss and intermediate losses are in the same order of magnitude for the initial epochs.

The training methodology for the supervised models is similar to E2E finetuning of the self-supervised models. The only difference being in E2E finetuning of models, we initialize the model weights with the SSL learned weights whereas for supervised models, we randomly intialized the weights.

To ensure that the improvement for our approach over the vanilla MoCo (baseline method) isn't due to different training times, we trained the baseline method to convergence (See Appendix C). The baseline method and our approach are pretrained for same number of epochs, and same batch size. These values are different for different datasets and are mentioned in section 4.2. The details of the data augmentations used for training the models are mentioned in Appendix E. Other hyper-parameters such as learning rate, embedding dimension, number of negative pairs, temperature scaling, encoder momentum, weight decay and sgd momentum are kept same for all the models. The value for the above hyper-parameters are as follows: learning rate = 0.3, embedding dimension = 128, number of negative pairs = 65536, temperature scaling = 0.07, encoder momentum = 0.99, weight decay = 1e-4 and sgd momentum = 0.9. We ran all our experiments on Nvidia Tesla V100 GPU with a memory of 32 GB. The code is available at `https://github.com/aakashrkaku/intermdiate_layer_matter_ssl`.

### 4.2 Datasets

#### 4.2.1 NIH chest x-ray dataset

NIH Chest X-ray dataset [Wang et al., 2017] has 112,120 x-ray images with disease labels from 30,805 unique patients. Authors use natural language processing and the associated radiological reports to text-mine disease labels. There are 14 disease labels: Atelectasis (10416 images), Cardiomegaly (2532 images), Effusion (12015 images), Infiltration (17852 images), Mass (5121 images), Nodule (5710 images), Pneumonia (1220 images), Pneumothorax (4794 images), Consolidation (4220 images), Edema (2103 images), Emphysema (2308 images), Fibrosis (1520 images), Pleural Thickening (3013 images) and Hernia (186 images), and one label for "No Findings" (54333 images) associated with each x-ray. We use full resolution images of size $1024 \times 1024$ to train models. The downstream task is to predict a binary label for each of the 14 disease labels for each x-ray. The data is split, by patient, into training (70%), validation (10%) and test (20%) sets. We evaluate the model performance using the macro average area under the receiver operating characteristics curve (AUROC) for 14 classes because the prior works [Wang et al., 2017, Seyyed-Kalantari et al., 2020] report the same. We also report the AUROC for each class for different self-supervised methods in Appendix A and Appendix B. We trained the models by all self supervised methods until the learning saturates which happens around 50 epochs with a batch size of 16.

#### 4.2.2 Diabetic retinopathy (DR) dataset

The EyePACS diabetic retinopathy [Voets et al., 2019b, Kaggle, 2015] dataset consists of 88,702 retinal fundus images. The training and validation set consists of 57,146 images with a 20% split for validation set, and the test set consists of 8790 images. We resize the images to $299 \times 299$ for training models. Each image has been rated by a clinician on a scale of 0-4, ranging from no diabetic retinopathy to proliferative diabetic retinopathy. Labels up to 1 are considered non-referable DR

(48,784 images) and 2-4 are considered referable DR (rDR) (17,152 images). The downstream task is a binary classification task to predict referable vs non-referable diabetic retinopathy. We evaluate the model performance using macro average area under the receiver operating characteristics (AUROC) curve for the 2 classes because the prior works [Voets et al., 2019a, Gulshan et al., 2016] working on this dataset report the same. We trained all the self supervised models until the learning saturates which happens around 100 epochs with a batch size of 32.

### 4.2.3   Breast cancer histopathology dataset

Invasive ductal carcinoma (IDC) is the most common form of breast cancer in women. The dataset [Janowczyk and Madabhushi, 2016, Cruz-Roa et al., 2014] consists of 277,524 patches (198,738 IDC negative and 78,786 IDC positive) of size $50 \times 50$ from 162 whole slide images of the tumor biopsies. We resize the images to $256 \times 256$ for training models. We use Bilinear interpolation to resize the $50 \times 50$ patch of breast cancer slides to $256 \times 256$. The downstream task is to predict a binary label for each patch. The data is split based on the whole slide images, into training (70%), validation (10%) and test (20%) sets. We evaluate the model performance using the F1-measure. F1-measure is the harmonic mean of precision and recall. The prior works [Janowczyk and Madabhushi, 2016, Cruz-Roa et al., 2014] working on this dataset also reported the F1-measure which makes the comparison easier. For readers' reference, we also report the AUROC score in the Appendix H. We trained all the self supervised models until the learning saturates which happens around 75 epochs with a batch size of 32.

## 4.3   Results

### 4.3.1   Performance for linear probing (LL fine-tuning) of models

Table 1 shows the results for linear probing of different self-supervised methods. We see that MoCo + MSE significantly outperforms other methods for diabetic retinopathy and breast cancer histopathology datasets. For NIH chest x-ray dataset, it is marginally better than a standard MoCo but not statistically significant. Table 1 also shows the performance of fully supervised models for reference.

Table 1: Performance of self-supervised methods for the downstream task after fine tuning only the linear classifier. For diabetic retinopathy and breast cancer histopathology datasets, MoCo + MSE significantly outperforms the standard MoCo. Whereas for NIH chest x-ray dataset, MoCo + MSE and MoCo lead to comparable performance.

| Dataset / Method | MoCo | MoCo + MSE | MoCo + Barlow Twins | Supervised |
|---|---|---|---|---|
| NIH Chest X-ray | 74.4 | **74.8** | 73.5 | 79.8 |
| (AUC (95% CI)) | (73.9-75.0) | **(74.2-75.4)** | (72.9-74.0) | (79.2-80.3) |
| Diabetic Retinopathy | 74.6 | **84.8** | 79.7 | 94.1 |
| (AUC (95% CI)) | (74.5-74.7) | **(84.6-85.0)** | (79.6-79.7) | (94.1-94.2) |
| Breast Cancer Histopathology | 80.7 | **82.5** | 82.3 | 82.7 |
| (F1-score (95% CI)) | (80.4-81.1) | **(82.2 - 82.9)** | (82.0-82.7) | (82.4-83.1) |

### 4.3.2   Performance for E2E fine tuning of models

Table 14 shows the results for E2E fine tuning of models trained via different self-supervised methods, fine tuned over different fractions of the labeled data (label fractions). All the models pretrained via self-supervised learning methods outperform the models trained via standard supervised learning, indicating that self-supervised learning provides a better initialization for the downstream classification tasks. For smaller labeled datasets (1% and 6% label fractions), models trained via MoCo + MSE and MoCo + Barlow twins significantly outperform the models trained via a standard MoCo. Compared to standard MoCo, our approach leads to an average performance gain of 5% and 2% for 1% and 6% label fractions, respectively, across three datasets. For NIH chest x-ray dataset trained with 100% label fraction, standard MoCo leads to marginally better performance than MoCo + MSE but this difference is not statistically significant.

Table 2: Performance of models trained via self-supervised learning methods in downstream classification tasks, after E2E fine tuning on different label fractions. For smaller labeled dataset (1% and 6% label fractions), MoCo + MSE and MoCo + Barlow twins lead to significantly better performance compared to standard MoCo. For 100% label fraction, except NIH chest x-ray dataset, MoCo + MSE and MoCo + Barlow twins lead to better performance compared to standard MoCo.

| Label fraction | MoCo | MoCo + MSE | MoCo + Barlow Twins | Supervised |
|---|---|---|---|---|
| NIH Chest X-ray (AUC (95% CI)) | | | | |
| 100% | **82.4 (81.7-83.0)** | 81.5 (80.9-82.1) | 80.0 (79.5-80.7) | 79.8 (79.2-80.3) |
| 6% | 69.8 (69.3-70.4) | **70.5 (69.9-71.0)** | 70.0 (69.2-70.6) | 65.2 (64.6-65.8) |
| 1% | 59.2 (58.6-59.9) | 61.4 (60.7-62.0) | **62.9 (62.3-63.5)** | 57.8 (57.2-58.4) |
| Diabetic Retinopathy (AUC (95% CI)) | | | | |
| 100% | 94.6 (94.3-94.6) | **96.6 (96.6-96.7)** | 95.7 (95.7-95.8) | 94.1 (94.1-94.2) |
| 6% | 92.4 (92.2-92.6) | **95.1 (94.8-95.2)** | 94.0 (94.0-94.3) | 69.1 (69.0-69.2) |
| 1% | 88.1 (88.1-88.4) | **93.6 (93.2-93.6)** | 92.5 (92.2-92.7) | 65.5 (65.4-65.6) |
| Breast Cancer Histopathology (F1-score (95% CI)) | | | | |
| 100% | 82.9 (82.6-83.3) | 85.7 (85.4-86.0) | **86.4 (86.1-86.7)** | 82.7 (82.4-83.1) |
| 6% | 82.8 (82.4-83.2) | **84.6 (84.2-84.9)** | 84.5 (84.2-84.8) | 82.7 (82.4-83.1) |
| 1% | 82.8 (82.5-83.2) | **85.1 (84.7-85.4)** | 84.4 (84.1-84.7) | 80.6 (80.3-81.0) |

## 4.4 Features analysis

### 4.4.1 Feature reuse

Each row of Table 3 shows the feature similarity between a pre-trained model and E2E fine tuned model for different intermediate layers. We perform this analysis for small labeled datasets because those are the datasets where we would like to leverage the features learned via self-supervision. In Table 3, we see that for almost all the blocks, models trained via MoCo+MSE and MoCo+Barlow twins have a higher feature reuse compared to the model trained via a standard MoCo. Additionally, the table shows that features from the earlier blocks are more similar before and after finetuning, compared to the features in the later blocks for all the self-supervised learning methods. This observation is consistent with Asano et al. [2019] which stated that self-supervised learning methods help to learn the low-level statistics of the data. These low-level image statistics are captured in the initial layers of the model.

Table 3: Feature Reuse: Feature similarity (measure using CKA) for different intermediate layers before and after E2E fine tuning of the models. Models trained via MoCo + MSE and MoCo + Barlow twins show higher feature similarity and higher performance than those trained via the standard MoCo. This suggests the features learned via MoCo + MSE and MoCo + Barlow twins are more reusable in various downstream tasks.

| | 1% labeled data | | | | | 6% labeled data | | | | |
|---|---|---|---|---|---|---|---|---|---|---|
| Method | Block 1 | Block 2 | Block 3 | Block 4 | Performance | Block 1 | Block 2 | Block 3 | Block 4 | Performance |
| NIH Chest X-ray (Performance in AUC) | | | | | | | | | | |
| MoCo | 0.81 | 0.80 | 0.57 | 0.41 | 59.2 | 0.72 | 0.75 | 0.66 | 0.39 | 69.8 |
| MoCo + MSE | 0.97 | 0.83 | 0.65 | **0.42** | 61.4 | **0.97** | 0.84 | 0.55 | **0.43** | 70.5 |
| MoCo + Barlow Twins | **0.99** | **0.98** | **0.76** | 0.38 | 62.9 | **0.97** | **0.92** | **0.79** | 0.41 | 70.0 |
| Diabetic Retinopathy (Performance in AUC) | | | | | | | | | | |
| MoCo | 0.87 | 0.80 | **0.51** | 0.19 | 88.1 | 0.81 | 0.69 | **0.50** | 0.14 | 92.4 |
| MoCo + MSE | 0.96 | 0.78 | 0.33 | **0.26** | 93.6 | **0.95** | 0.73 | 0.25 | 0.12 | 95.1 |
| MoCo + Barlow Twins | **0.98** | **0.83** | 0.58 | 0.24 | 92.5 | 0.93 | **0.75** | **0.50** | **0.15** | 94.0 |
| Breast Cancer Histopathology (Performance in F1-score) | | | | | | | | | | |
| MoCo | 0.50 | 0.55 | **0.98** | 0.16 | 82.8 | 0.57 | 0.48 | **0.97** | 0.19 | 82.8 |
| MoCo + MSE | **0.77** | **0.82** | 0.58 | **0.42** | 85.1 | 0.75 | **0.79** | 0.55 | 0.31 | 84.6 |
| MoCo + Barlow Twins | **0.77** | 0.74 | 0.54 | 0.36 | 84.4 | **0.76** | 0.76 | 0.58 | **0.38** | 84.5 |

### 4.4.2 Layer-wise probing

Table 4 shows the performance of models trained via different self-supervised learning methods when their intermediate features are used to perform the downstream task. In this analysis, we only train a linear classifier on the top of extracted intermediate features to perform the downstream tasks. Except for Block 1 of NIH chest x-ray, for rest of the blocks and datasets, the intermediate features of models trained via either MoCo+MSE or MoCo+Barlow twins outperform the model trained via a standard MoCo. This indicates that these models have better quality intermediate features.

Table 4: Layer-wise probing: Performance of models trained via self-supervised learning methods in the downstream classification tasks, using the intermediate features of the models. Training a model via MoCo + MSE and MoCo + Barlow twins leads to higher performance compared to standard MoCo, indicating that more informative features are learned via these methods.

|  | Block 1 | Block 2 | Block 3 | Block 4 |
|---|---|---|---|---|
| NIH Chest X-ray (AUC (95% CI)) | | | | |
| MoCo | **58.8 (58.4-59.3)** | 59.5 (59.0-60.0) | 65.3 (64.8-65.8) | 74.4 (73.9-75.0) |
| MoCo + MSE | 57.6 (56.9-58.3) | **59.9 (59.4-60.4)** | **69.2 (68.70-69.7)** | **74.8 (74.2-75.4)** |
| MoCo + Barlow Twins | 56.6 (56.2-57.0) | 56.5 (56.1-56.9) | 64.2 (63.7-64.6) | 73.5 (72.9-74.0) |
| Diabetic Retinopathy (AUC (95% CI)) | | | | |
| MoCo | 68.1 (68.0-68.1) | 68.2 (68.2-68.3) | 69.2 (69.2-69.5) | 74.6 (74.5-74.7) |
| MoCo + MSE | **68.3 (68.2-68.3)** | **70.1 (70.0-70.1)** | **71.2 (71.1-71.3)** | **84.8 (84.6-85.0)** |
| MoCo + Barlow Twins | 67.2 (67.2-67.3) | 68.6 (68.5-68.7) | 69.9 (69.4-69.9) | 79.7 (79.6-79.7) |
| Breast Cancer Histopathology (F1-score (95% CI)) | | | | |
| MoCo | 80.9 (80.5-81.3) | 81.1 (80.8-81.5) | 81.1 (80.7-81.5) | 80.7 (80.4-81.1) |
| MoCo + MSE | 80.6 (80.2-81.0) | **81.3 (81.0-81.7)** | **82.7 (82.4-83.0)** | **82.5 (82.2-82.9)** |
| MoCo + Barlow Twins | **81.0 (90.7-81.4)** | 81.1 (80.7-81.5) | 82.2 (81.9-82.6) | 82.3 (82.0-82.7) |

### 4.4.3 Probability comparison

Table 5 shows the KS distance between the output probabilities of E2E fine tuned models on small labeled datasets and the best performing models. The best performing models for different datasets are as follows: NIH chest x-ray dataset = a model trained via standard MoCo and fine tuned on 100% label fraction; Diabetic retinopathy dataset = a model trained via MoCo + MSE fine and tuned on 100% label fraction and breast cancer histopathology = a model trained via MoCo + Barlow twins fine tuned on 100% label fraction. For small labeled data, Table 5 shows that models trained via MoCo + MSE and MoCo + Barlow twins and E2E fine tuned on small data are closer to the best performing models compared to standard MoCo.

Table 5: Kolomongrov-Smrinov distance between the output probability distribution of the best performing models, and the models trained via self-supervised learning methods and E2E fine tuned on 1% or 6% label fractions. Except for NIH chest x-ray dataset with 6% label fraction, MoCo + MSE and MoCo + Barlow twins lead to models that are closer to the best performing model, compared to standard MoCo.

| Label fraction | MoCo | MoCo + MSE | MoCo + Barlow Twins | Supervised |
|---|---|---|---|---|
| NIH Chest X-ray (Compared to MoCo - Fine tuned on 100% labeled data) | | | | |
| 6% | **0.028** | 0.039 | 0.034 | 0.040 |
| 1% | 0.244 | **0.094** | 0.104 | 0.260 |
| Diabetic Retinopathy (Compared to MoCo + MSE - Fine tuned on 100% labeled data) | | | | |
| 6% | 0.21 | **0.12** | 0.30 | 0.37 |
| 1% | 0.31 | **0.18** | 0.22 | 0.60 |
| Breast Cancer Histopathology (Compared to MoCo + Barlow twins - Fine tuned on 100% labeled data) | | | | |
| 6% | 0.082 | 0.036 | **0.026** | 0.040 |
| 1% | 0.043 | 0.082 | **0.033** | 0.155 |

## 5 Discussion

The results from Table 1 and Table 14 show that the proposed approach of minimizing intermediate feature dissimilarity improves performance of the models for diverse downstream tasks and on diverse medical image datasets, as compared to a standard MoCo. The gains are even larger for a setting where we have a small labeled data (1% and 6% label fraction) to fine tune the models. The small labeled data settings are common in medical imaging domains, as expert manual effort is needed to annotate the images, leading to expensive and time-consuming labeling process. With this context, the gains achieved by our approach for small labeled datasets becomes more important.

Going beyond only assessing SSL in terms of predictive performance of downstream tasks, and using novel methods such as feature reuse analysis, layer-wise probing and probability distribution

comparisons, further helped us analyse the self-supervised learning methods, and quantify the quality of features learned via each self-supervision approach. Our finding suggests that better self supervised learning methods lead to objectively better features, in terms of reuse and informativeness.

**Limitations**: (1) Although we showed that the proposed approach improves performance for diverse datasets, due to computation resource constraints, we only showed it for one backbone architecture (ResNet-50) and one self-supervised baseline method (MoCo). But our approach of using intermediate layers to learn better features is simple and can be adapted to other backbone architectures and self-supervised learning methods. It would be interesting to investigate if our findings hold for other backbone architectures and self-supervised learning methods. (2) Our main interest was medical imaging, but our finding may also hold true in natural imaging domain. Due to computational constraints we did not test this method on ImageNet.

**Potential negative societal impact**: Our work primarily focuses on medical imaging. Artificial intelligence (AI) in medicine is still in an early stage. Hence, any commercial deployment of the tools/models presented in our work should not be done without sufficient evaluations. Proper FDA and other regulatory approval should be received to ensure appropriate use of the tools/models presented in our work.

## 6 Conclusion

We found that our proposed self-supervised learning method with the additional MSE or Barlow twins loss in the intermediate layers generates higher quality representations than the standard MoCo in three different medical imaging tasks. Our proposed methods also improved the performance of the models in downstream classification tasks, especially on low label fractions of the data. We evaluated the quality of the learned features by measuring the feature similarity before and after finetuning the model and probing the intermediate layers of the pre-trained model. The results from our proposed method indicates that the representations learned by the models are more reusable and informative compared to the standard MoCo. Finally, we showed that the models fine-tuned on low label fractions with the pretrained weights from our method get closer to the best performing model compared to standard MoCo, based on KS distance between output probability distributions.

To the best of our knowledge, this work is the first to show the importance of intermediate layers in momentum contrastive self-supervised learning and in a diverse set of medical imaging tasks. Despite the differences in the modalities of data, self-supervision via MoCo+MSE/Barlow twins leads to models that consistently outperformed or had similar performance on the downstream tasks.

## Acknowledgements

We would like to thank NYU Langone HPC team for assisting us with our computational needs. This work was supported by NYU Langone Health, Predictive Analytics Unit, and NYU Langone Radiology Department (NR) and NSF NRT-HDR Award 1922658 (AK).

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
