# A    NIH Chest X-ray: Performance for linear probing (LL fine-tuning) of models

Table 6 shows the performance of the different LL fine-tuned models on NIH Chest X-ray datatset for different classes.

Table 6: Area under ROC-curve of self-supervised methods for the downstream task after fine tuning only the linear classifier for NIH chest x-ray dataset. The last column shows the prevalence (in %) of each class in the test set. For most of the classes, either MoCo + MSE or MoCo + Barlow twins outperform the standard MoCo.

| Class | MoCo | MoCo+MSE | MoCo+Barlow Twins | Supervised on 100% labels | Prevalence (%) |
|---|---|---|---|---|---|
| Hernia | **78.64** | 76.89 | 76.77 | 86.90 | 0.19 |
| Pneumonia | **69.49** | 68.57 | 65.09 | 74.54 | 1.08 |
| Fibrosis | 70.73 | **73.21** | 71.90 | 76.40 | 1.61 |
| Edema | 85.01 | **85.23** | 84.04 | 87.62 | 1.84 |
| Emphysema | 75.70 | **76.38** | 73.71 | 85.87 | 2.27 |
| Cardiomegaly | 86.34 | **86.74** | 81.61 | 87.51 | 2.59 |
| Pleural Thickening | 70.49 | **71.55** | 69.25 | 73.13 | 3.27 |
| Consolidation | 75.56 | 76.14 | **76.33** | 78.49 | 4.27 |
| Pneumothorax | 76.42 | 77.36 | **77.50** | 82.92 | 4.85 |
| Mass | 69.90 | **70.21** | 69.70 | 79.42 | 5.05 |
| Nodule | 62.86 | 62.92 | **63.31** | 71.79 | 5.95 |
| Atelectasis | 73.69 | **74.29** | 74.09 | 77.71 | 10.79 |
| Effusion | **81.41** | 81.17 | 80.58 | 86.29 | 12.28 |
| Infiltration | 66.43 | **67.37** | 66.17 | 68.76 | 17.56 |

# B    NIH Chest X-ray: Performance for E2E fine tuning of models

Table 9, Table 8 and Table 7 shows the performance of the different E2E fine-tuned models on NIH Chest X-ray datatset for different classes.

## B.1    100% label fraction

Table 7: Area under ROC-curve of models trained via self-supervised learning methods in downstream classification tasks, after E2E fine tuning on 100% label fractions for NIH chest x-ray dataset. The last column shows the prevalence (in %) of each class in the test set. All the models pre-trained via self-supervised learning methods outperform the models trained via standard supervised learning, indicating that self-supervised learning provides a better initialization for the downstream classification tasks.

| Class | MoCo | MoCo+MSE | MoCo+Barlow Twins | Supervised | Prevalence |
|---|---|---|---|---|---|
| Hernia | **95.39** | 91.90 | 90.24 | 86.90 | 0.19 |
| Pneumonia | 74.81 | **74.92** | 73.81 | 74.54 | 1.08 |
| Fibrosis | **80.89** | 80.15 | 77.92 | 76.40 | 1.61 |
| Edema | 87.92 | 88.40 | **88.44** | 87.62 | 1.84 |
| Emphysema | **90.94** | 88.78 | 83.82 | 85.87 | 2.27 |
| Cardiomegaly | **90.94** | 90.66 | 90.88 | 87.51 | 2.59 |
| Pleural Thickening | **76.47** | 75.63 | 74.23 | 73.13 | 3.27 |
| Consolidation | **79.57** | 79.09 | 78.69 | 78.49 | 4.27 |
| Pneumothorax | **84.57** | 84.03 | 82.37 | 82.92 | 4.85 |
| Mass | **81.19** | 80.27 | 78.78 | 79.42 | 5.05 |
| Nodule | **73.74** | 72.28 | 69.31 | 71.79 | 5.95 |
| Atelectasis | **79.75** | 78.96 | 78.13 | 77.71 | 10.79 |
| Effusion | **87.75** | 87.04 | 85.96 | 86.29 | 12.28 |
| Infiltration | **69.57** | 69.14 | 68.47 | 68.76 | 17.56 |

## B.2  6% label fraction

Table 8: Area under ROC-curve of models trained via self-supervised learning methods in downstream classification tasks, after E2E fine tuning on 6% label fractions for NIH chest x-ray dataset. The last column shows the prevalence (in %) of each class in the test set. For most of the classes, either MoCo + MSE or MoCo + Barlow twins outperform the standard MoCo.

| Class | MoCo | MoCo+MSE | MoCo+Barlow Twins | Supervised on 6% labels | Supervised on 100% labels | Prevalence (%) |
|---|---|---|---|---|---|---|
| Hernia | 72.95 | **76.09** | 73.29 | 66.91 | 86.90 | 0.19 |
| Pneumonia | 64.92 | 64.51 | **67.39** | 63.14 | 74.54 | 1.08 |
| Fibrosis | **70.15** | 69.37 | 65.19 | 65.16 | 76.40 | 1.61 |
| Edema | 82.63 | 82.14 | **82.71** | 79.55 | 87.62 | 1.84 |
| Emphysema | **67.23** | 62.25 | 57.74 | 56.91 | 85.87 | 2.27 |
| Cardiomegaly | 64.47 | 74.80 | **80.05** | 65.84 | 87.51 | 2.59 |
| Pleural Thickening | **68.50** | 68.03 | 65.29 | 59.87 | 73.13 | 3.27 |
| Consolidation | 74.79 | 75.28 | **75.58** | 71.93 | 78.49 | 4.27 |
| Pneumothorax | **69.77** | 69.59 | 67.29 | 65.31 | 82.92 | 4.85 |
| Mass | 64.29 | 67.78 | **68.05** | 58.45 | 79.42 | 5.05 |
| Nodule | 61.43 | **61.74** | 59.42 | 54.74 | 71.79 | 5.95 |
| Atelectasis | 71.11 | 70.98 | **72.08** | 67.38 | 77.71 | 10.79 |
| Effusion | **81.07** | 79.41 | 80.12 | 76.13 | 86.29 | 12.28 |
| Infiltration | 65.08 | 65.00 | **65.08** | 61.52 | 68.76 | 17.56 |

## B.3  1% label fraction

Table 9: Area under ROC-curve of models trained via self-supervised learning methods in downstream classification tasks, after E2E fine tuning on 1% label fractions for NIH chest x-ray dataset. The last column shows the prevalence (in %) of each class in the test set. For most of the classes, either MoCo + MSE or MoCo + Barlow twins outperform the standard MoCo.

| Class | MoCo | MoCo+MSE | MoCo+Barlow Twins | Supervised on 6% labels | Supervised on 100% labels | Prevalence |
|---|---|---|---|---|---|---|
| Hernia | 50.02 | 58.21 | **69.85** | 59.14 | 86.90 | 0.19 |
| Pneumonia | 59.39 | 61.95 | **64.52** | 59.52 | 74.54 | 1.08 |
| Fibrosis | 56.39 | 55.34 | **60.07** | 53.64 | 76.40 | 1.61 |
| Edema | 72.96 | 74.51 | **79.36** | 75.66 | 87.62 | 1.84 |
| Emphysema | 51.31 | 56.53 | **57.89** | 49.00 | 85.87 | 2.27 |
| Cardiomegaly | **60.96** | 51.38 | 57.92 | 55.14 | 87.51 | 2.59 |
| Pleural Thickening | 55.57 | **61.35** | 55.89 | 52.56 | 73.13 | 3.27 |
| Consolidation | 69.01 | **70.40** | 64.76 | 65.39 | 78.49 | 4.27 |
| Pneumothorax | 55.23 | 58.37 | **60.42** | 56.45 | 82.92 | 4.85 |
| Mass | 55.21 | **61.86** | 60.76 | 53.81 | 79.42 | 5.05 |
| Nodule | 49.38 | 50.28 | **51.71** | 46.86 | 71.79 | 5.95 |
| Atelectasis | 63.95 | **64.41** | 63.95 | 58.92 | 77.71 | 10.79 |
| Effusion | 70.64 | **73.30** | 70.70 | 65.36 | 86.29 | 12.28 |
| Infiltration | 59.46 | 61.60 | **62.79** | 58.23 | 68.76 | 17.56 |

## C  Validation loss curves

To ensure that the improvement for our approach over the vanilla MoCo isn't due to different training times, we trained the vanilla MoCo to convergence. We considered the model to be converged when the validation loss during the self-supervised learning phase stops to improve. Figure 2 shows that the vanilla MoCo is trained to convergence.

## D  Class distribution for smaller datasets

In order to ensure that the smaller datasets which we use to train/fine tune the models are representative of the complete dataset, we report the class distribution of the different sizes for all three datasets in Table 10, Table 11 and Table 12.

## E  Augmentations used for training the models

We use the following augmentations for training all the methods (and for all datasets):

- Random color jittering with brightness, contrast, and saturation factor chosen uniformly from [0.6,1.4], and hue factor chosen uniformly from [-0.1,0.1]. This augmentation is applied with 80% probability.
- Random rotation between 0 to 30 degrees

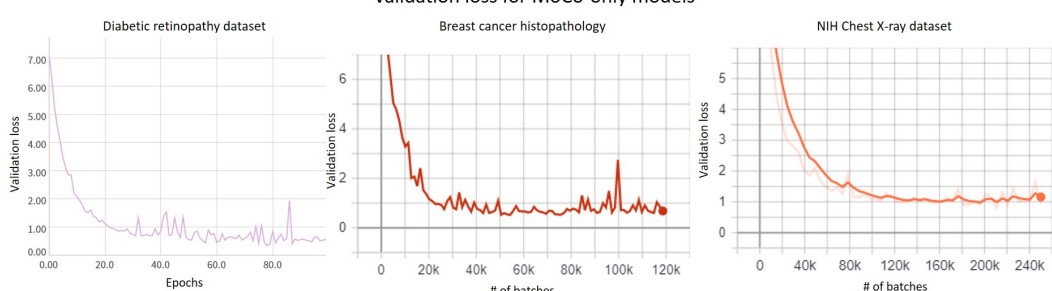

Figure 2: Validation loss for vanilla MoCo model for three datasets. The vanilla MoCo is trained to convergence which ensures that the improvement for our approach over the vanilla MoCo isn't due to different training times

Table 10: Class distribution for different sizes of NIH-Chest Xray dataset. The class distribution of smaller datasets match the class distribution of the complete dataset. Thereby, we ensure class distribution does not change when we downsample the dataset.

| NIH Chest X-ray | Fraction of positives in train set | | |
|---|---|---|---|
| Classes | 100% | 6% | 1% |
| Atelectasis | 0.1019 | 0.1025 | 0.1046 |
| Cardiomegaly | 0.0249 | 0.0257 | 0.0204 |
| Effusion | 0.1180 | 0.1025 | 0.1033 |
| Infiltration | 0.1773 | 0.1584 | 0.1607 |
| Mass | 0.0508 | 0.0480 | 0.0497 |
| Nodule | 0.0558 | 0.0544 | 0.0408 |
| Pneumonia | 0.0125 | 0.0106 | 0.0077 |
| Pneumothorax | 0.0472 | 0.0340 | 0.0778 |
| Consolidation | 0.0416 | 0.0429 | 0.0434 |
| Edema | 0.0215 | 0.0202 | 0.0179 |
| Emphysema | 0.0229 | 0.0236 | 0.0702 |
| Fibrosis | 0.0148 | 0.0187 | 0.0204 |
| Pleural Thickening | 0.0290 | 0.0236 | 0.0281 |
| Hernia | 0.0018 | 0.0013 | 0.0013 |

- Random Gaussian blurring with sigma chosen with uniform distribution between 0.1 and 2.0
- Random gray scaling with 20% probability

We also horizontally flipped images with 50% probability for the breast histopathology and diabetic retinopathy datasets.

## F    Ablation study for intermediate loss

We performed a preliminary ablation analysis with one of the dataset, NIH-Chest Xray dataset, to understand to which blocks of ResNet-50 should we apply the intermediate loss. Table 13 shows the outcome of this ablation analysis. Based on the preliminary analysis, it was seen that applying

Table 11: Class distribution for different sizes of Diabetic Retinopathy dataset. The class distribution of smaller datasets match the class distribution of the complete dataset. Thereby, we ensure class distribution does not change when we downsample the dataset.

| Diabetic Retinopathy | Fraction of positives in train set | | |
|---|---|---|---|
| | 100% | 6% | 1% |
| % of referrable DR | 0.2884 | 0.2901 | 0.3019 |

Table 12: Class distribution for different sizes of Breast Cancer Histopathology. The class distribution of smaller datasets match the class distribution of the complete dataset. Thereby, we ensure class distribution does not change when we downsample the dataset.

| Breast Cancer Histopathology | Fraction of positives in train set | | |
|---|---|---|---|
| | 100% | 6% | 1% |
| % of positive cases | 0.2811 | 0.2730 | 0.284 |

intermediate loss to all blocks yielded superior results. We, therefore, choose to apply intermediate loss to all the blocks for all our experiments.

Table 13: Ablation study for intermediate loss applied to different blocks of the ResNet encoder (for NIH-Chest Xray dataset). The table show the performance of the models when fully fine-tuned on 100% of labeled training set. The preliminary ablation study gave the evidence that applying intermediate loss to all blocks yielded superior results. Here, blocks "1-2-3-4" signifies that the intermediate loss was applied to blocks 1, 2, 3 and 4.

| | Intermediate loss (blocks) | AUC | MoCo (AUC) | Supervised (AUC) |
|---|---|---|---|---|
| MoCo + MSE | **1-2-3-4** | **81.5** | 82.4 | 79.8 |
| | 2-3-4 | 79.5 | | |
| | 3-4 | 80.8 | | |
| | 4 | 80.1 | | |
| MoCo + Btwins | **1-2-3-4** | **80** | | |
| | 2-3-4 | 78.9 | | |
| | 3-4 | 79.9 | | |
| | 4 | 78.7 | | |

## G  Procedure to compute KS distance between two models

We first sort the logit outputs of the model for all the data points in test set in ascending order and then divide them into 40 discrete bins. We normalize the counts of each bin by the total number of data points. The normalized counts obtained for each bin act as the discretize pdf which is used to build an empirical CDF. We build this CDF for both the models under comparison. Then, we use the sci-py's implementation of "ks_2samp" to compute the KS distance between the two CDFs.

## H  Results for Breast Cancer Histopathology dataset in AUC

Table 14: AUC Performance of models trained via self-supervised learning methods in downstream classification tasks, after LL and E2E fine tuning on different label fractions. For all fraction of labeled dataset (1%, 6% and 100% label fractions), MoCo + MSE and MoCo + Barlow twins lead to significantly better performance compared to standard MoCo, in terms of AUC.

| Label fraction | MoCo | MoCo + MSE | MoCo + Barlow Twins | Supervised |
|---|---|---|---|---|
| | Breast Cancer Histopathology (AUC (95% CI)) | | | |
| 100% LL | 79.5 (79.2-80.0) | 82.2 (81.8-82.6) | **82.8 (82.5-83.3)** | - |
| 100% E2E | 82.5 (82.1-82.9) | 84.3 (83.9-84.6) | **86.4 (86.1-86.7)** | 81.6 (81.3-82.0) |
| 6% E2E | 81.3 (80.9-81.7) | **84.6 (84.2-85)** | 84.1 (83.7-84.4) | 81.9 (81.5-82.3) |
| 1% E2E | 82.0 (81.6-82.4) | **86.4 (86.1-86.7)** | 83.5 (83.1-83.8) | 79.4 (79.0-79.8) |