# OpenReview forum: "Intermediate Layers Matter in Momentum Contrastive Self Supervised Learning"
_NeurIPS.cc/2021/Conference — NeurIPS 2021 Poster_

### Official Review · Reviewer_aDCN · 2021-07-12

**Rating:** 6
**Confidence:** 3

**Summary:**

The authors propose an augmentation to MoCo whereby they augment the standard InfoNCE loss with an additional loss that encourages the intermediate layers of a ResNet50 to be close for multiple views of the same image. They demonstrate the efficacy of this loss on several medical imaging classification tasks.

**Limitations And Societal Impact:**

The authors have adequately addressed the societal impact and limitations of their work.

**Main Review:**


Originality:
The idea is a natural extension to a popular technique in a very trendy field. The loss functions used in the intermediate layer are either the first-choice when encouraging similarity (MSE), or a recent development that has been shown to work well (BT). However the use of an MLP projection for the BT loss or the average pooling for MSE are nontrivial and indicate that the authors have had to work a little and not simply do the most-naive thing for these losses.

Quality:
The emphasis on purely medical imaging tasks is fairly nonstandard and makes the comparison of the proposed approach to other techniques challenging. When comparing against vanilla MoCo, the intermediate layer losses provide a boost in performance in almost all cases, and surprisingly outperform the fully supervised setting in certain cases. However, I am slightly skeptical about the comparison to the baseline. Intuitively, the training times of models trained with the intermediate-loss will be larger than the vanilla MoCo training times (on a per-epoch basis), and there is no assertion that the vanilla MoCo was trained to maximum convergence. An alternative hypothesis that could explain these results (and thus needs to be ruled out) could be that vanilla MoCo is able to attain the same performance as the proposed methods, but requires more epochs of training; the different training times per epoch, however, could mean that vanilla MoCo can attain this equivalent performance using the same wall-clock time. I would like to see training curves, some notion of wall-clock time, and some argument that the proposed methods actually do outperform MoCo regardless of training duration/hyperparameter tuning. Another way to ameliorate my concern would be to demonstrate the performance of the MoCo+X on datasets more commonly used in the SSL literature. Further, comparisons against other SSL techniques should be performed: is MoCo+MSE better than {SWaV, Barlow Twins, etc...} on these tasks (ignoring the benifits that Moco+X is doesn't require the overhead of negative examples)?

Significance:
Learning models that are able to extract useful features without labels is very important, and is doubly so in instances where labeled data is hard to obtain (such as in medical applications). While this work seems like a natural training trick without strong theoretical backing, this trick is applicable to many different self-supervised learning schemes and seems to improve the performance with very limited overhead or hyperparameter tuning, and may even provide more useful features.

Clarity:
The paper is well-written and easy to follow.

Overall: I think this paper is marginally above the acceptance threshold, as a cute and easy improvement to self-supervised learning techniques that seems to improve performance in most cases. I'm hesitant to push for a full-accept because of the experimental concerns I pointed out in the 'quality section', and the emphasis on nonstandard datasets that makes a thorough comparison to SOTA hard to tease out.

**Time Spent Reviewing:**

3

---

> ### Author Response · Authors · 2021-08-10
> **Thank you so much for carefully reviewing the manuscript and giving us valuable feedback. We have tried to address your comments/questions below in as much detail as possible.**
>
> Thank you so much for carefully reviewing the manuscript and giving us valuable feedback. Please find below our responses to your comments:
> 1. The wall-clock time/epoch for training our approach (MoCo+MSE and MoCo+BT) and a vanilla MoCo model are almost the same. To ensure that the improvement for our approach over the vanilla MoCo wasn’t due to different training times, we trained the vanilla MoCo to convergence. We considered the model to be converged when the validation loss during the self-supervised learning phase stops to improve. The plot for validation loss for three datasets can be seen [here](https://imgur.com/28FmYta). We will include the training curves in the appendix of the camera-ready version.
>
> 2. Another way to ameliorate the quality concerns would be to compare and show that the performance of our supervised models (on 100% dataset) is better than or comparable to the ones reported in the prior similar works. For example, for the NIH chest x-ray dataset, our supervised model (as well as SSL models) outperforms the model reported [here](https://openaccess.thecvf.com/content_cvpr_2017/papers/Wang_ChestX-ray8_Hospital-Scale_Chest_CVPR_2017_paper.pdf). Similarly, for DR dataset, the performance of the supervised model achieves similar performance to the one reported [here](https://www.ncbi.nlm.nih.gov/pmc/articles/PMC6553744/pdf/pone.0217541.pdf) whereas the SSL models outperform it. Finally, for the breast histopathology dataset, our supervised model (as well as the SSL models) outperforms the models’ performance reported in [this](https://spie.org/Publications/Proceedings/Paper/10.1117/12.2043872?SSO=1) and [this](https://pubmed.ncbi.nlm.nih.gov/27563488/).
>
> 3. Comparisons with other methods: It would be desirable to compare the performance of our approach with other SSL methods like Swav, BYOL, etc. But we focussed the paper on MoCo because outside the natural image domain, especially in the medical domain, MoCo is widely used but never explored enough. We chose to dig deeper into MoCo in our work. In future studies, we will focus on expanding our analysis to other SSL methods as well.

---

### Official Review · Reviewer_FLzm · 2021-07-15

**Rating:** 6
**Confidence:** 4

**Summary:**

This paper examines the utility of minimizing the distance between the values of intermediate layers in a neural network model, when performing supervised learning with the momentum contrastive (MoCo) method on two augmented versions of the same image. Two metrics for minimizing said distance are tested: mean squared error, and a cross-correlation matrix by the Barlow twins loss function (Zbontar et al., 2021). Experiments on three medical image datasets show that enforcing intermediate layer closeness can produce improve classification performance, especially when the proportion of labelled data for training is small. Additional experiments suggest that intermediate layer features trained in this way are more suited for reuse for various downstream tasks, and are of higher quality.

**Limitations And Societal Impact:**

Yes.

**Main Review:**

1. It might be clarified whether the intermediate feature similarity enforcements apply only to positive pairs, and the standard MoCo for negative pairs (in their respective encoder) remains unchanged; if so, it might be commented as to whether intermediate features may be exploited for negative pairs too.
2. It is stated that Resnet-50 is the backbone architecture for all baseline self-supervised methods (apparently MoCo) and our approach(es) (+MSE/+Barlow Twins) (Line 222). The architecture and training methodology for the supervised baseline might be stated as well, and justified to be a fair comparison.
3. For linear probing (LL fine-tuning) as reported in Section 4.3.1/Table 1, what was the label fraction used, and is there any reason why various label fraction values were not investigated, as was done for end-to-end (E2E) fine-tuning? It might also be considered to standardize the format of Table 1 with Table 2 (e.g. “Supervised” column to come first)
4. The class distributions for the 1% and 6% label fraction experiments might be stated. In particular, are the original class distributions maintained? This is of interest as medical image datasets often have highly imbalanced class distributions, and rare classes may thus possibly not be represented when sampling for low label fractions. Ideally, the class label distributions/quantities for each dataset might be provided, for training/validation/test splits, possibly in the supplementary Appendix.
5. It might be interesting to investigate saliency heatmaps for the various methods (supervised, MoCo, MoCo+MSE, MoCo+Barlow Twins) if possible, to further verify whether intermediate layer similarity affects classification attribution.
6. Minor issues:
(Line 62) “have high performance” might be “has high performance”
(Line 86) “We don’t” might be “We do not”
(Line 236) “The” is repeated.
(Line 259) “proliferate diabetic retinopathy” might be “proliferative diabetic retinopathy”
---
We thank the authors for addressing our major concerns in their reply.

**Time Spent Reviewing:**

2

---

> ### Author Response · Authors · 2021-08-10
> **Thank you so much for carefully reviewing the manuscript and giving us valuable feedback. We have tried to address your comments/questions below in as much detail as possible.**
>
> Thank you so much for carefully reviewing the manuscript and giving us valuable feedback. Please find below our responses to your comments:
> 1. Intermediate feature similarity applies only to positive pairs. We thank the reviewer for pointing this out. We will clarify the same in the camera-ready version.
>
> 2. Supervised model architecture and training: We use exactly the same backbone architecture (Resnet-50) for the supervised models. The training methodology for the supervised models is similar to E2E finetuning of the self-supervised models (the only difference being in E2E finetuning of models, we initialize the model weights with the SSL learned weights). We will make these points clear in the camera-ready version.
>
> 3. LL for smaller datasets: This could be a question for the reader, therefore we will add it to the appendix. However, we chose to exclude it because the models trained using 1% or 6% of data and only training the last linear layer, have low performance. We won’t get useful insights by comparing low-performing models. At such low-performance levels, the differences between the models are not statistically significant.
>
> 4. Class distribution for smaller datasets: The class distributions of smaller distributions match the class distribution of the complete dataset (please find the class distributions [here](https://imgur.com/OE28mUa)). We will make it clear in the camera-ready version.
>
> 5. Saliency maps: This is an interesting idea. However, objectively evaluating saliency maps is challenging ([paper1](https://arxiv.org/pdf/1511.01844.pdf)) and prior work ([paper2](https://papers.nips.cc/paper/2018/file/294a8ed24b1ad22ec2e7efea049b8737-Paper.pdf))  also shows that many SOTA saliency maps suffer from confirmation bias. Therefore we avoid saliency maps and focus on quantitative methods like feature reuse and layer-wise probing, to analyze the proposed methods.
>
> 6. Thank you for pointing out the minor typos. We will fix them in the camera-ready version.

---

### Official Review · Reviewer_ZiVQ · 2021-07-15

**Rating:** 7
**Confidence:** 4

**Summary:**

The authors propose to learn augmentation invariant features early on in a network by, in addition to a global contrastive loss, also optimizing for an intermediate loss function to ensure similarity between features in early layers of the network. They test two loss functions for this purpose - MSE and a minimization of the difference between the cross-correlation matrix of the intermediate representations and the identity matrix. They hypothesize that this way the network learns high quality features early on.
The authors also test the quality of features learnt with their method in a self-supervised setting, by comparing the similarity of the features learnt under self-supervision to features learnt after fine-tuning on a labeled dataset following the self-supervision. They do so for a variety of different labelled set sizes, as well as end-to-end fine-tuning, and only tuning the last layer.

They test their method on three publicly available medical datasets (chest X-Ray, diabetic retinopathy and histopathological whole slide images), which are of different imaging modalities.


**Limitations And Societal Impact:**

The authors did not discuss the negative societal impact in sufficient detail. Their method is in no way limited to medical images, i.e. they should reflect upon other data in which self-supervision is key.


**Main Review:**

The paper is well structured and well written. The idea is clearly presented, as well as the means how to evaluate the feature quality. The claims are supported by experiments on three available datasets of different modalities which shows that the versatility of the proposed idea to regularize intermediate feature representations by an additional similarity loss term. The concept can be used for any backbone architecture and enforce various invariances of the representations depending on the data augmentation. Seeing how the method shows major improvements in classification performance over the baseline in case of only small labelled datasets, this work could be very valuable, for biomedical tasks in particular.



There are however a number of questions to address in the rebuttal:



It should be stated more clearly in the methodology section how the InfoNCE loss is combined with the loss on the intermediate layers.

I think it is very relevant to point out precisely what augmentations were used during the self-supervision pretraining (on top of citing Chen et al) as they will lead to the representations’ invariances.

Assuming rotations were part of the augmentation, I wonder if the drop in performance of the proposed method vs only MoCo is due to the orientation of the data in case of the NIH Chest X-Ray dataset? The histopathological images are rotationally and translationally invariant by nature, the diabetic retinopathy images to some extend also, if the meta data available to the dataset (left/right eye, inversion of the photo or not) was not used to correct for it (which is how I understand the dataset was used here for both the MoCo experiments as well as for the proposed method). However, the chest X-Ray images have a set orientation, so learning rotationally invariant features could possibly be counterproductive. Similarly, invariances to color distortion could prohibit the network to pick up on darker or lighter regions in the X-Ray which may be crucial for classification.  Hence my suggestion (possibly for future work) would be to adjust the augmentation such that only invariances are learnt that are in line with the clinical background knowledge available and see if the performance can be further improved that way.

Why were the sizes, in percent, of the training/test/validation sets chosen differently for the DR dataset than the other two datasets?

What interpolation method was used for the resizing? Seeing how the patches from the whole slide images increased its pixel count by more than 26x, resized from 50x50 to 256x256, the interpolation method will play a significant role in the resulting input images.

Why was AUC reported for the DR binary classification task, but the F1-measure for the histopathology dataset? I strongly suggest to report both values for both datasets to make comparison of the improvement of the proposed method feasible.



Minor remarks:
It would be good to point out in the caption of Table 3 that CKA was used as a similarity measure.

It would be beneficial to adjust Fig. 1 to the same notation used in Eqn. (1) (views of the same image denoted as $\tilde{x_i}$ in the equation, but as $v$ and $v’$ in the figure for example).

It is unclear to me from the paper what part the “momentum” plays portrayed in Fig. 1.

Some minor spelling/grammar mistakes, mostly with singular/plural or capitalization (lines 17, 23, 34, 38, 68, 158, 166, 180, 201, 204, 222, 228, 240, 247, 248, 278, other instances of X-ray)
Some issues in the citations, e.g Neyshabur et al 2021 missing journal/proceedings

-------------------------------------------------------------------------------------------------------------------
After rebuttal:
The authors promise to include details which I was missing in the submission, in the camera-ready version and I higher my score to 7.

**Time Spent Reviewing:**

8

---

> ### Author Response · Authors · 2021-08-10
> **Thank you so much for carefully reviewing the manuscript and giving us valuable feedback. We have tried to address your comments/questions below in as much detail as possible.**
>
> Thank you so much for carefully reviewing the manuscript and giving us valuable feedback. Please find below our responses to your comments:
> 1. Combining InfoNCE and MSE/BT loss: The InfoNCE loss is added to the four intermediate feature losses from each block of the Resnet-50 encoder using a scaling factor. The scaling factor multiplies the intermediate feature loss by 0.25 (in case of MSE loss) or by 5e-5 (in case of BT loss). We chose these scaling factors so that the contrastive loss and intermediate losses are in the same order of magnitude for the initial epochs. We will make this more clear in the methodology section of the camera-ready version.
>
> 2. Augmentations: We use the following augmentations for training all the methods: color jittering, rotation (only between 0 to 30 degrees), gaussian blurring, and gray scaling. We also use Horizontal flipping for the Breast Cancer and DR dataset. We will explicitly mention these augmentations with exact parameters in the appendix section of the camera-ready version.
>
> 3. Effect of augmentation/relevant augmentations: For the NIH chest x-ray dataset, we did not use horizontal flipping instead we used color jittering, rotation (only between 0 to 30 degrees), gaussian blurring, and gray scaling. These are valid augmentations for a chest x-ray because due to patient positioning there are variations in the chest x-rays, and chest x-ray data is not typically registered to a reference. We use the exact same augmentations and augmentation parameters for all the methods (vanilla MoCo, proposed method, and supervised models). Therefore, the performance comparison between different methods is fair and valid.
>
> 4. Train-val-test size of DR dataset: Since we were replicating results for the baseline model from [[1](https://www.ncbi.nlm.nih.gov/pmc/articles/PMC6553744/pdf/pone.0217541.pdf)] - which replicates the experiments from [[2](https://jamanetwork.com/journals/jama/fullarticle/2588763)] using the Kaggle EyePacs data, we used the same splits as mentioned in the above paper. For other datasets (Chest X-ray and Histopathology dataset), we use the standard split of 70-10-20 for the train-val-test.
>
> 5. Interpolation methods: We use Bilinear interpolation to resize the 50x50 patch of breast cancer slides to 256 x 256. We will add this detail to the camera-ready version of the paper.
>
> 6. AUC vs F1: Since the prior works [[3](https://openaccess.thecvf.com/content_cvpr_2017/papers/Wang_ChestX-ray8_Hospital-Scale_Chest_CVPR_2017_paper.pdf), [4](https://arxiv.org/pdf/2003.00827v2.pdf)] report the average AUC for the NIH chest x-ray dataset, we also report the same to ensure the readers can compare the performance of our models with prior works. Similarly, for the DR dataset, prior works like [[5](https://www.ncbi.nlm.nih.gov/pmc/articles/PMC6553744/pdf/pone.0217541.pdf)] and [[6](https://jamanetwork.com/journals/jama/fullarticle/2588763)] report the AUC scores. Therefore, we also report the same. For the breast histopathology dataset, the prior work [[7](https://spie.org/Publications/Proceedings/Paper/10.1117/12.2043872?SSO=1), [8](https://pubmed.ncbi.nlm.nih.gov/27563488/)] reports the F1 score, therefore to ensure we have comparable or better performance as compared to the prior works, we report an F1 score. But, we will also report the AUC and F1 score for DR and histopathology dataset in the appendix.
>
> 7. Minor remarks:
>
> a. We will make it clear that we use CKA as the measure of similarity in the feature reuse table (Table 3).
>
> b. Thank you for pointing this out. We will update the figure so that it matches the notations that we used in the equations.
>
> c. Role of “momentum”. The “momentum” in the figure denotes that the weights of the second encoder are an exponential moving average of the weights of the first encoder. We will make it explicit in the image caption for better clarity.
>
> d. Typos and grammatical errors: Thank you for pointing out the errors. We will fix them in the camera-ready version.

---

> > ### Comment · Reviewer_ZiVQ · 2021-08-16
> > **Addressing the Author Response**
> >
> > I thank the authors for addressing my concerns in the initial review in detail.
> > The authors promise to include details which I was missing in the submission, in the camera-ready version.
> >
> > I am considering increasing my score based on the rebuttal but will wait if there is going to be any further discussion among the other reviewers, authors and chairs here.

---

### Official Review · Reviewer_vuAC · 2021-07-21

**Rating:** 5
**Confidence:** 4

**Summary:**

This paper improves the self-supervised contrastive learning framework, specifically MoCo, by making the features in the intermediate layers closer. Although not the first attempt at this idea, this paper dig to investigate the choice of loss terms and the property of learned features. MSE loss and a recently represented Barlow Twins loss are studied. For analyzing learned features, layer-wise probing measures the performance of using intermediate features for downstream tasks (higher means better). Feature reuse is measured by quantifying the similarity of features between models after pertaining, finetuning on labeled data. The idea is that the features change less over finetuning means self-supervised learning learns better features. The output distributions of modes finetuned with different amounts of labels are also compared with the best-performed model as a metric for finetuning with limited labeled data. This paper focuses on medical images in the study.

**Limitations And Societal Impact:**

Yes.

**Main Review:**

Strength:
- This paper explores a nature idea to extend the current contrasting frameworks by adding intermediate features. Two loss terms used in the paper don't need negative pairs, which saves the computation cost. Both losses improve the original MoCo in linear eval and e2e fine-tuning except the NIH chest X-ray dataset and MSE loss works better than Balow twins in more cases.
- They analyze learned features from a few interesting perspectives and may inspire other research.

Weakness/Concerns/Suggestions:
- It is unclear why 3.4.1 "feature reuse" and 3.4.3 "Feature utility for small datasets choose" different comparison targets: 3.4.1 choose the 100% labels finetuned model for each method while 3.4.3 compare to the best-performed model. I am a bit concerned about the choice of 3.4.3. The distance may not only be related to the performance but also the approach itself. In other words, is it really true that the probability similarity with the best model is strongly correlated with the actual performance? There are counter-cases e.g. in 6% labeled  Breast cancer data, the supervised model has a similar distance with MoCo-MSE and is better than MoCO while it has the worst accuracy. Maybe checking the correlation between the distances and the actual performance is helpful to answer this question.

- It will be helpful if the paper can discuss more (clearly) on that what properties the three analysis can provide about regarding the feature quality that the downstream task performance cannot. In other words, why do people working on self-supervised representation learning want to do these analyses on their approaches?

- A ablation study on applying the intermediate feature similarity to not all positions is missing.

- Some details are missing:
  - Why use KS distance but not other metrics e.g. KL divergence?
  - How is the similarity/distance computed. My best guess is the average of the per-sample results over the test set.
  - section 3.3, how do you use 1000 bootstrap replicates to measure the performance?

Overall, I think this paper presents a reasonable approach to extend MoCo that shows good performance. The other contribution is the three ways of analyzing feature quality. However, authors should highlight/discuss more their value as general metrics for evaluating learned representation if they are not only used for the approach in this paper.  Consider the limited scope of studied model/backbone/datasets (discussed in the conclusion though), I am slightly lean to reject this paper.



**Time Spent Reviewing:**

5

---

> ### Author Response · Authors · 2021-08-10
> **Thank you so much for carefully reviewing the manuscript and giving us valuable feedback. We have tried to address your comments/questions below in as much detail as possible.**
>
> Thank you so much for carefully reviewing the manuscript and giving us valuable feedback. Please find below our responses to your comments:
> 1. Feature reuse: Intuitively, if two models pre-trained via self-supervised learning reach a similar performance after fine-tuning on a downstream task, the SSL method that leads to the model with features closer to the fine-tuned model has learned better/more-useful features during the SSL stage. Therefore, for feature reuse, we compare the feature similarity between the SSL-pretrained model and the E2E finetuned model.
> Whereas, for “Feature utility for small datasets” (section 3.4.3), we wanted to see how close the models fine-tuned on smaller datasets can get to those that are fine-tuned on large datasets. Therefore, we compare models trained via our SSL methods, fine-tuned on 1% and 6% of the labeled training data, and the best model fine-tuned on 100% of the labeled training data. For most cases, KS distance is a good indication of the performance of the models. But, in some cases, as pointed out by the reviewer, KS distance may be high (bad) but performance is still good. This would be the case when the models under comparison are differently calibrated. In other words, low KS distance would mean classification performance, and the calibration of the model trained on a smaller dataset matches well with the model trained on the larger dataset. Whereas, high KS distance can mean two things, 1: bad model performance (and calibration), or 2: just bad calibration. Since this point is important for interpreting the KS distance, we will make it clear in the camera-ready version.
>
> 2. Benefits of feature analyses in addition to the downstream task performance: Typically, in the medical domain [[1](https://arxiv.org/pdf/2011.13971.pdf),[2](https://arxiv.org/pdf/2012.03583v1.pdf),[3](https://arxiv.org/pdf/2010.05352.pdf),[4](https://arxiv.org/pdf/2101.05224.pdf)], the models pretrained with SSL are fully fine-tuned. In such scenarios, just from the downstream task performance, it is unclear how much did the SSL stage and the fine-tuning stage contribute to the downstream task performance. To answer this question, analyses like feature reuse and layer-wise probing can be useful to evaluate the quality of features learned during the SSL phase.
>
> 3. Ablation study for int. Feature similarity: Given the limited computation resource and time, we were not able to complete the ablation study for intermediate features. We will include it in the appendix of the camera-ready version.
>
> 4. Missing details:
>
> a. KS vs KL: KS is a standard metric used in the statistics community to ascertain the closeness of two distributions with statistical significance. Additionally, KS distance is symmetrical, unlike KL divergence. Therefore, to avoid the problem of asymmetry when reporting the results, we use KS distance.
>
> b. How is agg. KS distance computed? The individual points help us to plot a CDF of the output probabilities of the model. The reported KS distance is computed by measuring the maximum separation between the CDF of the two models (under comparison).
>
> c. How bootstrap replicates are used? We compute the performance measure for each bootstrap replicate. Using the 1000 replicates, we report a 95% confidence interval. We compute the confidence interval using μ±1.96σ/√N where, μ and σ are mean and standard deviation of the performance measure for 1000 bootstrap replicates, respectively.

---

> > ### Comment · Reviewer_vuAC · 2021-09-02
> > **Review updates after rebuttal**
> >
> > I appreciate the authors' efforts in addressing my concerns. After reading the authors' rebuttal and other reviewer's feedback. I decide not to change my rating.
> > - I understand the goal of measuring feature reuse and feature utility is to separate the contribution of the SSL/finetuning stage. But I disagree that we cannot tell from performance only. This is typically done by looking into performance under linear-eval (layer-wise probing) and using less labeled data for fine-tuning. Generally, the better performance achieved after fine-tuning with the same amount of labeled data, the better representation SSL learned. Unless the labeled data is sufficient to get good performance after finetuning regardless of the quality of SSL representations.
> > - Authors claims "higher feature reuse indicates better feature quality" referring to Neyshabur et al. 2021. However, in Neyshabur et al. 2021, feature reuse was studied in transfer learning and the CKA metric is computed as evidence that pre-trained weights do transfer useful features instead of just low-level image statistics when compared with randomly initialized weights. Neyshabur et al. did not simply conclude that higher feature reuse indicates better feature quality for comparing different pretrained weights. It is a hypothesis in this submission and the authors did not show enough support to this hypothesis.
> > - The presented setup of feature utility for small datasets makes the observations trivial. All methods + label fractions are compared to the same best model. It is natural that the models from the same training method tend to generate similar output distributions, which is true in most cases. For example, for Diabetic Retinopathy data, the MOCO+MSE models are closer to the best model that is also a MOCO+MSE model.
> > - I understand that KS-distance is computed with CDF. But what exactly the distribution of model output means? Did you compute the CDFs of each of the output logits? If this metric has been firstly used to compare two metrics, more details should be included to help readers to understand.
> > - I also agree with other reviewers' concerns on other aspects of the paper's quality e.g. the evaluation and comparison of the design is not thorough and some details are missing. Although this paper focuses on medical domain data, missing all general vision benchmarks hurts the quality of the paper as a general SSL approach.

---

> > > ### Author Response · Authors · 2021-09-12
> > > **Thank you for reconsidering the review rating**
> > >
> > > We thank the reviewer for reconsidering the rating. Please find below our responses to your comments:
> > >
> > > 1. As mentioned in the above response, typically in the medical domain, the models are fully finetuned after the SSL stage. One of the reasons for doing so is because only finetuning the last layer is not enough to get a model that has good enough performance as compared to a model trained in a supervised manner from scratch. This is unlike the Imagenet dataset where only finetuning the last layer almost reaches the supervised model’s performance. So, in such a setting, since the final model ends up being the one that is fully finetuned after the SSL stage, it would be better to use feature reuse and layer-wise probing to evaluate the contribution of the SSL phase and finetuning phase.
> > >
> > > 2. As mentioned in the above point, in the medical domain, the models are typically fully finetuned after the SSL stage. Therefore, this setting is akin to transfer learning where the pre-training is happening during the SSL stage. In our work, we, therefore, leveraged the methods like feature reuse to evaluate the SSL pre-training phase. Intuitively, if two models pre-trained via self-supervised learning reach a similar performance after fine-tuning on a downstream task, the SSL method that leads to the model with features closer to the fine-tuned model has learned better/more-useful features during the self-supervised learning. Here, it is important to note that just a higher feature similarity of a model doesn’t signify it is better. We should also ensure that it has similar or better performance on the downstream task as compared to other models. That is the reason, in table 3, we also show the performance on the downstream task.
> > >
> > > 3. For Diabetic Retinopathy data, the closeness between MOCO+MSE models to the best model is because they better replicate the performance of the best model. On the other hand, for the Chest X-ray dataset, the closest model to the best model (which is a MOCO model) is not a MOCO model but MOCO+MSE and MOCO+BT models because they have better replicate the performance of the best MOCO model. Therefore, the statement "it is natural that the model from the same training methods tends to generate similar output distributions" has counter-examples.
> > >
> > > 4. We first sort the logit outputs of the model for all the data points in ascending order and then divide them into discrete bins. We normalize the counts of each bin by the total number of data points. The normalized counts of each bin act as the discretize pdf which is used to build an empirical CDF. We thank the reviewer for pointing this out and will include the exact procedure in the appendix of the camera-ready version.
> > >
> > > 5. It would be desirable to compare the performance of our approach with other SSL methods like Swav, BYOL, etc. But we focussed the paper on MoCo because outside the natural image domain, especially in the medical domain, MoCo is widely used but never explored enough. We chose to dig deeper into MoCo in our work. We also focused on designing ways to evaluate the SSL feature representation beyond the downstream performance. In future studies, we will focus on expanding our analysis to other SSL methods as well.

---

### Decision · Program_Chairs · 2021-09-27

**Decision:**

Accept (Poster)

**Comment:**

This paper proposes a straight-forward but clever and useful extension of MoCo by adding intermediate features.
A similar approach has been proposed earlier (Gong et al., withdrawn from ICLR, on openreview, cited in the proposed paper) this paper makes a great case for it.  The paper tests on three very different, biomedical and openly available datasets and achieve very good results. They have addressed and promised to include details which the reviewers requested from the original submission so overall this paper was viewed positively by all reviewers and meta-reviewer.